# In Vitro Cultured *Melissa officinalis* Cells as Effective Ingredient to Protect Skin against Oxidative Stress, Blue Light, and Infrared Irradiations Damages

**Giovanna Pressi** [1,*], **Oriana Bertaiola** [1], **Chiara Guarnerio** [1], **Elisa Barbieri** [1,*], **Flavia Guzzo** [2], **Caroline Durand** [3], **Laurent Peno-Mazzarino** [3], **Veronica Cocetta** [4], **Isabella Giacomini** [4] and **Alessandra Semenzato** [5]

1    Demethra Biotech S.R.L., 36043 Camisano Vicentino (VI), Italy; orianabertaiola@dembiotech.it (O.B.); chiaraguarnerio@dembiotech.it (C.G.)
2    Department of Biotechnology, University of Verona, 37134 Verona, Italy; flavia.guzzo@univr.it
3    Laboratoire BIO-EC, 91160 Longjumeau, France; c.durand@bio-ec.fr (C.D.); l.peno-mazzarino@bio-ec.fr (L.P.-M.)
4    Department of Pharmaceutical and Pharmacological Sciences, University of Padova, 35131 Padova, Italy; veronica.cocetta@unipd.it (V.C.); isabella.giacomini@phd.unipd.it (I.G.)
5    Unired S.R.L, 35131 Padova, Italy; alessandra.semenzato@unired.it
*    Correspondence: giovannapressi@dembiotech.it (G.P.); elisabarbieri@dembiotech.it (E.B.)

**Abstract:** Skin is being increasingly exposed to artificial blue light due to the extensive use of electronic devices, which can induce cell oxidative stress, causing signs of early photo aging. The *Melissa officinalis* phytocomplex is a new standardized cosmetic ingredient obtained by an in vitro plant cell culture with a high content of rosmarinic acid. In this study, we examine the activity of the *Melissa officinalis* phytocomplex to protect skin against blue light and infrared damages, evaluating the ROS (Radical Oxygen Species) level in keratinocyte cell line from human skin (HaCaT) and Nrf2 (Nuclear factor erythroid 2-related factor 2), elastin, and MMP1 (Matrix Metalloproteinase 1) immunostaining in living human skin explants ex vivo. This phytocomplex demonstrates antioxidant activity by reducing ROS production and thus the oxidant damage of the skin caused by UV and blue light exposure. In addition, it inhibits blue light-induced Nrf2 transcriptional activity, IR-induced elastin alteration, and IR-induced MMP-1 release. This *Melissa officinalis* phytocomplex is a new innovative active ingredient for cosmetic products that is able to protect skin against light and screen exposure damages and oxidative stress.

**Keywords:** *Melissa officinalis*; in vitro cell culture; rosmarinic acid; phytocomplex; oxidative stress; blue light



## 1. Introduction

*Melissa officinalis* L. (lemon balm) is an edible perennial herb of the Lamiaceae family. Its name originates from the Greek words for bee (*melissa*) and honey (*meli*). There are worldwide records of its medicinal and culinary use that date back to Dioscorides (father of pharmacology) times (40–90 A.D.) and allow its safe exploitation [1]. *Melissa officinalis* is reputed in folk medicine for memory-enhancing effects, promoting long life, action against gastrointestinal disorders, rheumatism, Alzheimer's, thyroid diseases, colic, anemia, nausea, vertigo, syncope, asthma, bronchitis, amenorrhea, cardiac disorders, epilepsy, insomnia, migraines, nervousness, malaise, depression, psychosis, hysteria, and wounds [2]. Several scientific papers confirm the medicinal effectiveness of *Melissa officinalis* preparations, as well as its antioxidant and other properties suggesting its use for the prevention of oxidative stress-related diseases [3]. The bioactivity of *Melissa officinalis* extracts is mainly attributed, as for any other plant formulation, to the qualitative and quantitative composition of secondary metabolites (i.e., phenolic acids, flavonoids, and terpenoids). Rosmarinic acid is a caffeic acid ester with 3,4-dihydroxyphenyllactic acid,

and it is a main bioactive component of the *Melissa officinalis* extracts [1]. The rosmarinic acid and phenolic acids content in the *Melissa officinalis* extract is highly variable. The variability is associated with multiple factors, which are difficult to control: seasons, plant age, geographical growing areas, and tissues used for the preparation of products [4]. Furthermore, the preparation of standardized *Melissa officinalis* derivatives with a reproducible content of metabolites is very difficult. The extreme variability in the content of phytoconstituents of plant preparations obtained directly from a plant, or parts thereof, by extraction negatively impacts the effectiveness of the same. An alternative method for obtaining contaminant-free standardized plant preparation in industrial quantities is to use in vitro cell cultures.

Plant cell culture technology is a technique for growing plant cells under strictly controlled environmental conditions [5] that makes it possible to solve the problems tied to the variability of plant extracts, since it provides preparations with a content of active substances that can be reproduced in a standardized manner. The derived extracts can be easily standardized in their primary and secondary metabolites and are compliant with the safety requirements of being contamination-free and phytochemically uniform [6]. In addition, to respect biodiversity and greater environmental sustainability with a drastic reduction in the use of natural resources such as water and soil, this technology makes it possible to overcome seasonal and geographical limitations by guaranteeing a higher safety profile for the consumer (free from heavy metals, pesticides, aflatoxins, bacterial or fungal contamination) and an elevated degree of standardization. The objective of the present study concerns the appraisal of the biological activity of a product rich in rosmarinic acid obtained by a stable and selected cell line of *Melissa officinalis* [7]. In this work, we demonstrate the efficacy of a *Melissa officinalis* product obtained by in vitro cell cultures to protect the skin against blue light and IR damages.

The skin is the main defensive barrier in the body against a large variety of environmental factors and is responsible for maintaining body homeostasis, defence, and repair [8]. It is well known that exposure to internal and external factors activates lots of molecular processes that damage skin structure. The sunlight spectrum consists of short, high-energy wavelengths, from ultraviolet radiation (280–400 nm) to visible light (400–700 nm) and infrared radiation (700 nm–1 mm), long and low energy wavelengths [9]. Visible light wavelengths penetrate the deepest parts of the dermis, reaching the different types of skin cells and inducing cell oxidative stress, damage, and dysfunction, causing signs of early photo-aging [10]. Skin is being increasingly exposed to artificial blue light due to the extensive use of electronic devices. This, together with recent observations reporting that blue light can exert cytotoxic effects associated with oxidative stress and promote hyperpigmentation, has sparked interest in blue light and its potential harmful effects on skin [11]. Several studies report that blue light contributes to skin aging and carcinogenesis, mostly during direct sunlight exposure, similar to UVA [12].

Numerous pharmacological activities of rosmarinic acid have been described. Among other activities, it protects cells membranes against oxidative damage [13]. Fernando et al. [14] showed that rosmarinic acid treatment of keratinocyte cell line from human skin (HaCaT) cells damaged from UVB radiation recovered the expression levels of Nrf2 (Nuclear factor erythroid 2-related factor 2) protein from the cytosol into the nucleus. These results indicate that rosmarinic acid may protect cellular environments from free-radical damage and thereby enhance the cellular antioxidant defense system. In this work, we examine the efficacy of an ingredient highly standardized in rosmarinic acid that was obtained by a selected cell line of *Melissa officinalis*, to protect the skin against blue light and IR damages. Especially, we demonstrated the activity of this new ingredient in countering the harmful action of ROS (Radical Oxygen Species), in inhibiting the expression of Nrf2 and MMP1 (Matrix Metalloproteinase 1), and in protecting the degradation of elastin. The evaluation of these parameters has allowed us to ascertain that the *Melissa officinalis* phytocomplex obtained by in vitro cell culture, standardized in rosmarinic acid, is able to reduce skin damages caused by ROS, blue light, and IR exposure.

## 2. Materials and Methods

### 2.1. Melissa officinalis Cell Culture

Cell culture of *Melissa officinalis* L. was obtained from seeds bought and certified from the nursery plant "Le Georgiche", Brescia, Italy.

The sterile plants of *Melissa officinalis* L. were obtained from seed sanitized by means of a treatment in sequence with 70% (*v/v*) ethanol (Honeywell, Wunstorfer Straβe 40, D-30926 Seelze, Germany)) in water for about 2 min, 2% (*v/v*) of sodium hypochlorite solution (6–14% active chlorine, (MERCK KGaA, 64271 Darmstadt, Germany) and 0.1% (*v/v*) Tween 20 (Duchefa, Postbus 809, 2003 RV-Haarlem, The Netherlands) for 15 min and, finally, at least 4 washes with sterile distilled water. The sanitized seeds were placed in trays containing nutrient medium Gamborg B5 [15] rendered solid by adding 0.8% *w/v* of plant agar (Duchefa) and without growth hormones. After a suitable period of incubation under light (15 days) and at 25 °C, the seeds began to sprout. Twenty days after germination, small leaves were collected from the plants grown under sterile conditions. The leaves were fragmented into small pieces (explants) of sub-centimetric dimensions (0.1–0.5 cm). The fragments of plant tissue were deposited in Petri dishes containing solidified Gamborg B5 medium supplemented with 20 g/L sucrose (Sudzucker AG, Manheim, Germany), 1 mg/L of (NAA) naphtalenacetic acid (Duchefa), 1 mg/L of (IAA) indoleacetic acid (Duchefa), 0.5 mg/L of (K) Kinetin (Duchefa), and 0.9% (*w/v*) of plant agar, final pH 6.5 (MO Medium). The pH was adjusted to 6.5 before autoclaving. Explants were incubated at 25 °C in the dark. Calli grown on *Melissa officinalis* phytocomplex (MO) medium were subjected to subculture for at least 4 months until they became friable and homogeneous, with a constant growth rate (*Melissa officinalis* stable cell line). The botanical origin of *Melissa officinalis L.* was confirmed by fingerprint DNA analysis made from Parco Tecnologico Padano, Lodi, Italy [16].

A part of selected calli (10% *w/v*) was transferred into 250 ml of culture liquid medium (MO medium without Plant Agar) contained in Erlenmeyer flasks of 1 liter volume. The suspension cultures were maintained at 25 °C in the dark in constant agitation on the rotary shaker at 120 rpm, and every 7 days of fermentation, they were subjected to a liquid subculture. The volume of biomass was increased by subculture on flasks of 3l volume containing 1l of fresh liquid medium culture. The amount of suspension culture inoculated into the liquid medium was equal to 6% *v/v*.

To increase the content of rosmarinic acid, after 7 days of fermentation, the suspension culture was inoculated in a bioreactor of 5L volume containing 3L of MO final liquid medium (Gamborg B5 with the addition of 35 g/L of sucrose, 0.5 mg/L di NAA, 0.5 mg/L of IAA, and 0.25 mg/L of K, final pH 6.5). The suspension culture was grown for a culture cycle of 14 days.

### 2.2. Phytocomplex Preparation from Melissa officinalis Cell Culture

After 14 days of growth, at 25 °C and in the dark, the *Melissa officinalis* cell suspensions were filtered by 50 μm mesh filter, and the medium cultures were discarded. Cells were washed with a double volume of saline solution (0.9% *w/v* NaCl in sterile water) and added with 1% (*w/w*) of citric acid and then homogenized with ultraturrax at 15,000 rpm for 20 min. The biomass of homogenated cells was dried to obtain a powder of phytocomplex with a high content of rosmarinic acid.

### 2.3. Quantification of Rosmarinic Acid and the Total Polyphenols in the Melissa officinalis Phytocomplex by UPLC-DAD

First, 100 mg of powder of the *Melissa officinalis* phytocomplex were weighed into a 15 ml test tube, and 30 volumes of ethanol (Honeywell) and water 60:40 (*v/v*) were added. The suspension was mixed for 30 seconds with a vortex mixer and sonicated for 15 min in an ice bath; finally, it was centrifuged at 4000 rpm for 15 min at 6 °C. At the end of centrifugation, the supernatant was recovered. Then, 15 mL of supernatant were transferred into a new test tube 15 and preserved in ice until loading into the UPLC

system. The sample was diluted 1:10 (first 1:5 in a solvent and then 1:2 in water). The diluted sample was filtered over 0.22 μm filters before being loaded into the UPLC system. Five independent replicates of the phytocomplex were extracted and measured. The chromatography system used for the quantification of rosmarinic acid consists of an Acquity UPLC BEH C18 1.7 μm column, size 2.1 mm × 100 mm, coupled to an Acquity UPLC BEH C18 1.7 μm VanGuard Pre-Column 3/Pk, size 2.1 mm × 5 mm. The platform used for the UPLC-DAD (Ultra Performance Liquid Chromatography-Diode Array Detection) analysis comprises a UPLC system (Waters Corporation, Milford, MA 01757, USA) consisting of an eluent management module, Binary Solvent Manager model I Class, and an auto-sampler, Sample Manager—FTN model I Class, coupled to a PDA (Photo Diode Array) eλ diode array detector. Empower 3 (Waters) software was used to acquire and analyze the data. The chromatography method used was the following: solvent A: water, 0.1% formic acid; solvent B: 100% acetonitrile. The initial condition is 99% solvent A; moreover, the flow remains constant at 0.350 mL/min throughout the duration of the analysis. The chromatography column was temperature controlled at 30 °C. Elution of the molecules was conducted by alternating gradient and isocratic phases, as indicated in Table 1.

**Table 1.** Elution of the molecules in UPLC-DAD analysis.

| Time from Start of the Analysis (Minutes) | Percentage of Solvent B | Slope |
| :---: | :---: | :---: |
| 0 | 1% | |
| 1 | 1% | linear |
| 11 | 40% | linear |
| 12 | 100% | linear |
| 13 | 100% | linear |
| 13.10 | 1% | linear |
| 15 | 1% | linear |

For quantification of the rosmarinic acid, the chromatogram obtained at the wavelength of 330 nm was used. The rosmarinic acid was quantified thanks to the calibration curve of the authentic commercial standard of rosmarinic acid (CAS 20283-92-5; purity≥95%; Sigma Aldrich). The data analysis was carried out with Empower 3 software.

*2.4. Evaluation of ROS Levels in Immortalized Keratinocyte Cell Line from Human Skin (HaCat)*

ROS were quantified using diacetylated 2′,7′-dichlorofluorescein (DCF-DA) probe (Sigma-Aldrich, Merck, Darmstadt, Germany) in HaCaT cells. HaCaT cells were grown in high-glucose Dulbecco's modified Eagle's media (DMEM) supplemented with 10% Fetal Bovine Serum (FBS), 2% L-glutamine, and 1% penicillin/streptomycin. Cells were maintained at 37 °C under a humidified atmosphere of 5% $CO_2$ in air.

HaCaT cells ($5 \times 10^3$) were seeded into 96-well plates, allowed to adhere overnight, and then treated for 24 h with the compounds, according to the experimental protocol. ROS levels were measured after the addition of 100 μM DCF-DA, further incubation for 30 min at 37 °C, and wash with phosphate-buffered saline (PBS) [17] as previously described. N-acetyl-L-cysteine 5 mM (Sigma Aldrich) and Rutin at 0.1% *w/v* were used as positive controls and the *Melissa officinalis* phytocomplex was used at 0.1% *w/v*.

Probe fluorescence intensity was measured at excitation 485 nm–emission 535 nm, in the absence or presence of 1.1 μM $H_2O_2$ used as oxidative stimulus, using a Multilabel Plate Reader VICTOR X3 (PerkinElmer). Fold increases in ROS production were calculated using the equation:

$$(F_{treatment}—F_{blank})/(F_{control}—F_{blank}) \tag{1}$$

where F is the fluorescence reading.

The statistical analysis was performed using GraphPad Prism version 6 for Windows (GraphPad Software, San Diego, CA, USA). Results are presented as mean ± SD of 3 experiments. An unpaired Student's t-test was used to compare ROS values, and *p* values <0.05 were considered statistically significant.

### 2.5. Assessment of Protection Activity of Melissa Officinalis Phythocomplex Against Irradiations of Infrared and Blue Light on Living Human Skin Explants Ex Vivo

#### 2.5.1. Sample Preparation and Explant Distribution

Human living skin explants were prepared from an abdominoplasty (aesthetic surgery) from a 35-year-old healthy Caucasian woman with a phototype II-III (Fitzpatrick classification), after obtaining informed consent. Adipose tissue was removed, and explants of $1\ cm^2$ were prepared using a circular biopsy punch. The explants were maintained in survival cell culture conditions at 37 °C in a humid atmosphere enriched with 5% $CO_2$ in BIO-EC's explant medium. BIOEC's explant medium is a proprietary culture medium specifically engineered by the BIO-EC Laboratory for the survival of skin explants.

#### 2.5.2. Product Preparation and Application

The *Melissa officinalis* phytocomplex (MO) has been prepared in sterile distilled water at 0.05% and 0.1% (*w/v*). Successively, the MO at 0.05% and MO at 0.1% were applied topically based on $2\ mg/cm^2$ and spread using a small spatula on day 0 (D0), day 1, day 4, and day 5 (30 min before blue light or infrared exposure). The control explants "C" did not receive any treatment.

#### 2.5.3. Blue Light Irradiations

Skin explants from the batches "BL", "MO at 0.05% + BL", and "MO at 0.1% + BL" were exposed to blue light using the Solarbox®device (SATIE, Cergy-Pontoise, France), which is a visible light radiation system, designed and produced by the SATIE laboratory (CNRS joint research unit UMR 8029, Cergy Paris University Cergy-Pontoise, France). The light source is ensured by a set of configurable light-emitting diodes able to deliver visible blue radiation with an emission spectrum of 420 to 580 nm, with a peak at 455 nm.

On day 5, the explants of the batches mentioned above were exposed to a blue light dose of $63.75\ J/cm^2$. The unirradiated explants were kept in the dark during the whole time of blue light exposure.

#### 2.5.4. Infrared Irradiation

Skin explants from the batches "IR", "MO at 0.05% + IR", and "MO at 0.1% + IR" were irradiated using an infrared lamp (Dr FISCHER 1000W, 235V 2500K; 760–1150 nm) on day 5, with an infrared dose of $720\ J/cm^2$. The unirradiated explants were kept in the dark during the whole time of infrared exposure.

#### 2.5.5. Histological Processing

On day 6 (D6), 24 h after blue light or infrared exposure, the explants from each batch were collected and cut in two parts. Half was fixed in buffered formalin, and the other half was frozen at −80 °C.

After fixation in buffered formalin, the samples were dehydrated and impregnated in paraffin using a Leica PEARL dehydration automat. The samples were embedded using a Leica EG 1160 embedding station. Then, 5-μm-thick sections were made using a Leica RM 2125 Minot-type microtome, and the sections were mounted on Superfrost®(Menzel-Gläser, Braunschweig, Germany) histological glass slides.

#### 2.5.6. Staining and Immunostaining

The cell viability of epidermal and dermal structures was observed on formol-fixed paraffine-embedded (FFPE) skin sections after Masson's trichrome staining, Goldner variant. The cellular viability was assessed by microscopical observation. Both immunohistochemistry and immunofluorescence stainings were performed on either 5μm-thick FFPE skin sections or 7μm-thick frozen sections using the following primary antibody: anti-phosphorylated Nrf2 antibody (Abcam, Cambridge, UK, ref. ab76026, clone EP1809Y), anti-elastin antibody (Novotec, Bron, France, ref. 25011), anti-MMP-1 antibody (Sigma, Merck KGaA, Darmstadt, Germany, ref. M4696). For immunohistochemistry, a ready-to-use

biotinylated universal secondary antibody has been used (Vector Laboratories, Burlingame, CA, USA), combined with a streptavidin–peroxidase amplifying system and revealed by the VIP chromogen (Vector Laboratories, Burlingame, CA, USA). Elastin immunostaining was revealed by immunofluorescence using an Alexa Fluor®488- conjugated secondary antibody (Life technologies, Thermo Fisher Scientific, Carlsbad, CA, USA, A11008). Nuclei were counterstained with propidium iodide (Sigma-Merck KGaA, Darmstadt, Germany).

The microscopical observations were realized using an Olympus BX63 microscope. Pictures were digitized with a numeric DP74 Olympus camera with cellSens Dimension storing software (Olympus, Tokyo, Japan). The intensity level of each immunostaining was scored from 1 to 6.

### 2.6. Statistical Analysis

The statistical analysis was performed using GraphPad Prism version 3.03 for Windows (GraphPad Software, San Diego, CA, USA). Results are presented as mean $\pm$ SD of 3 experiments. An unpaired Student's t-test was used to compare ROS values and, $p$ values <0.05 were considered statistically significant.

### 3. Results

#### 3.1. Melissa officinalis Phytocomplex With High Content of Rosmarinic Acid Was Obtained by a Stable and Selected Cell Line

A stable and selected cell line of *Melissa officinalis* was obtained using the MO solid medium. In this selected solid medium, the *Melissa officinalis* cell line is beige-colored with brown tinges and has a friable texture and a high rate of growth (subculture in fresh solid medium every 14 days). Figure 1 shows the cell culture maintained in solid MO medium (a) and cells of the line Mo-4AR seen under an AXIO-Imager A2 optical microscope (ZEISS), in the bright field mode (b) and after staining with fluorescein diacetate (c).

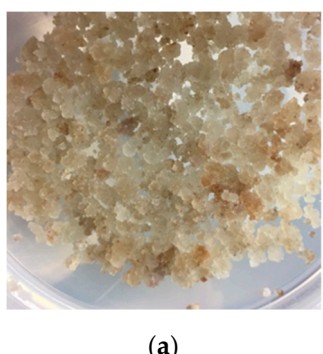 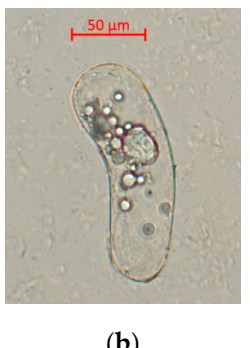 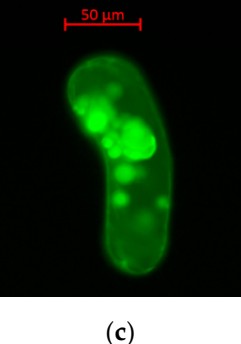

(**a**)        (**b**)        (**c**)

**Figure 1.** *Melissa officinalis* cell culture maintained in solid MO medium: (**a**) cells of the *Melissa officinalis* seen under an AXIO-Imager A2 optical microscope (ZEISS), in the bright field mode (**b**) and after staining with fluorescein diacetate (**c**).

The content of rosmarinic acid and total polyphenols in the cell line was optimized using a MO final liquid culture medium with a higher content of sucrose (35g/L) and with a lower concentration of growth hormones (0.5 mg/L of NAA, 0.5 mg/L of IAA, and 0.25 mg/L of K). The cells grown into MO final liquid culture medium were used to prepare the *Melissa officinalis* phytocomplex. The UPLC-DAD analysis was used to estimate the content of rosmarinic acid and total polyphenols content into the *Melissa officinalis* phytocomplex. The chromatographic profile of the phytocomplex at 330 nm is shown in Figure 2. The data obtained show that the content of rosmarinic acid in the phytocomplex is $7.6 \pm 0.1\%$ $w/w$ and the content of total polyphenols expressed as equivalent in rosmarinic acid is $9.2 \pm 0.1\%$ $w/w$. Thus, the main component of total polyphenols in the phytocomplex is represented by rosmarinic acid (83% of the total polyphenols).

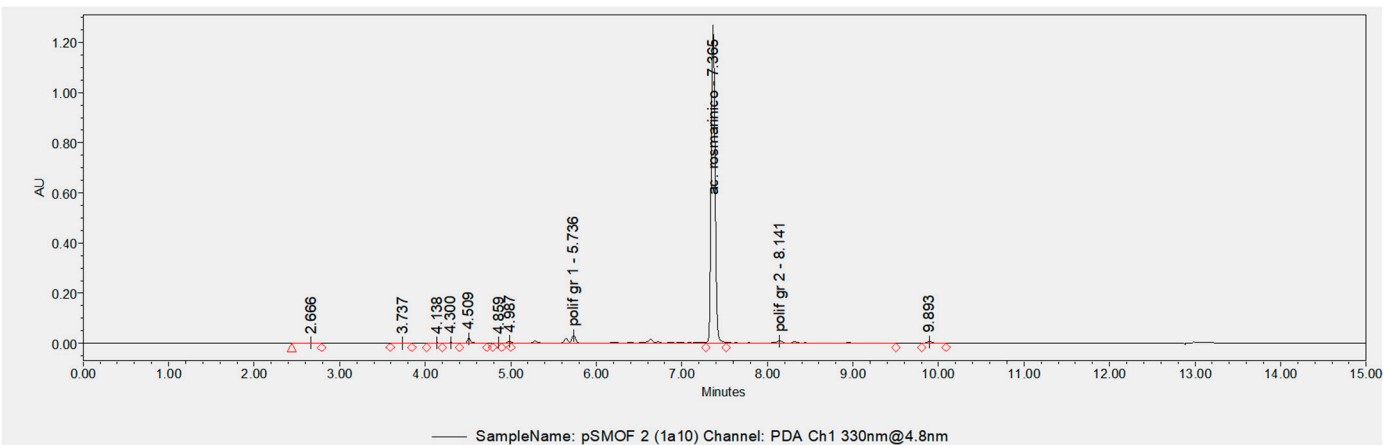

**Figure 2.** Chromatographic profile of the *Melissa officinalis* phytocomplex. The main peak at retention time of 7.3 min corresponds to rosmarinic acid.

### 3.2. Melissa officinalis Phytocomplex Exhibits Strong Inhibition of ROS Production after $H_2O_2$ Oxidative Stimulus in HaCaT Cell Line

In order to evaluate the ability of the *Melissa officinalis* phytocomplex to modulate the oxidative stress, ROS levels in HaCaT cells were evaluated, following 24 h' treatment. ROS generation was evaluated both in basal conditions and following the oxidative stimulus, which was induced by the addition of $H_2O_2$. The obtained results showed a significant decrease in ROS production in HaCaT cells treated with the *Melissa officinalis* phytocomplex, in the presence of oxidative stimulus, able to increase ROS levels, as shown in the untreated cells (Figure 3). The positive antioxidant controls, NAC 5 mM and Rutin 0.1% $w/v$, confirmed the anti-oxidant effect of the *Melissa officinalis* phytocomplex.

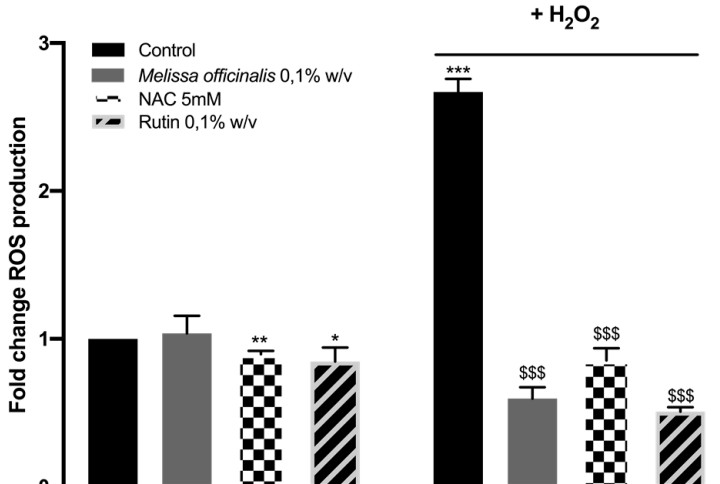

**Figure 3.** ROS (Radical Oxygen Species) levels in HaCaT cells after 24 h treatment with the *Melissa officinalis* phytocomplex 0.1% $w/v$, NAC (N-Acetyl-L-Cysteine) 5 mM and Rutin 0.1% $w/v$. ROS were detected by diacetylated 2′,7′-dichlorofluorescein (DCF-DA) staining at basal condition and following the exposure to oxidative stimulus $H_2O_2$. The fluorescence was measured by using a Victor3X multilabel plate counter (Ex 485 nm and Em 535 nm). Results are expressed as fold change of ROS production compared to control. Each bar represents the mean $\pm$ SD of n = 3 experiments. * $p < 0.05$, ** $p < 0.01$, *** $p < 0.001$, treatment vs. basal control. $^{\$\$\$}$ $p < 0.001$, treatment vs. stimulated control.

### 3.3. Melissa officinalis Phytocomplex Is Active to Protect the Human Skin Explants against Irradiations of Infrared and Blue Light Damages

### 3.3.1. Cell Viability

The application of the *Melissa officinalis* phytocomplex at 0.05% $w/v$ and at 0.1% $w/v$ on human skin explants induces no modification on the cell viability compared to the untreated control on D6, demonstrating that the product is well tolerated by ex vivo human skin.

Neither the blue light nor infrared irradiation induces modification of the cell viability. Upon irradiations, the application of the *Melissa officinalis* phytocomplex at 0.05% and at 0.1% $w/v$ induces no modification of the cell viability to the batches (Figure 4).

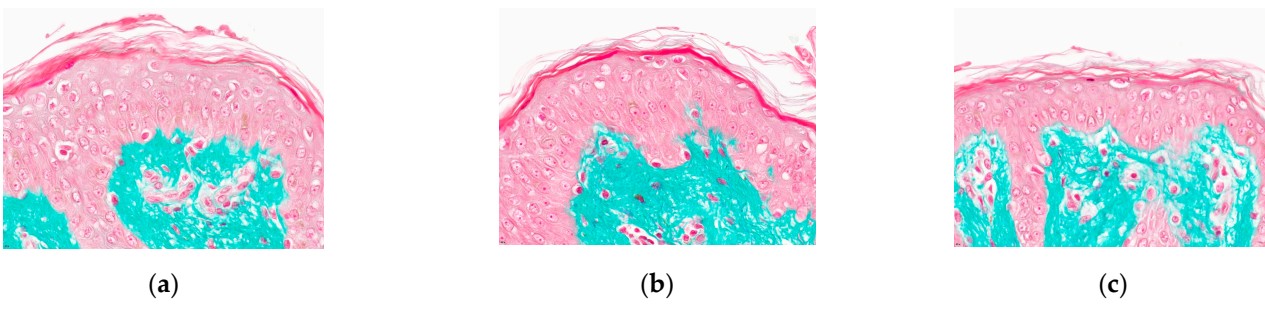

(**a**)        (**b**)        (**c**)

**Figure 4.** Cell viability of the untreated control on day 6 (**a**), of the batch treated with 0.05% $w/w$ of the *Melissa officinalis* phytocomplex on day 6 (**b**) and of the batch treated with 0.1% $w/w$ of *Melissa officinalis* phytocomplex on day 6 (**c**).

### 3.3.2. Nrf2 Immunostaining

The results of the immunostaining using the antibody anti-phosphorylated Nrf2 in the living epidermidis are summarized in Figure 5. After 6 days of treatment, the *Melissa officinalis* phytocomplex 0.05% (P1) or 0.1% $w/v$ (P2) does not modify the basal level of Nrf2, compared to the untreated control batch.

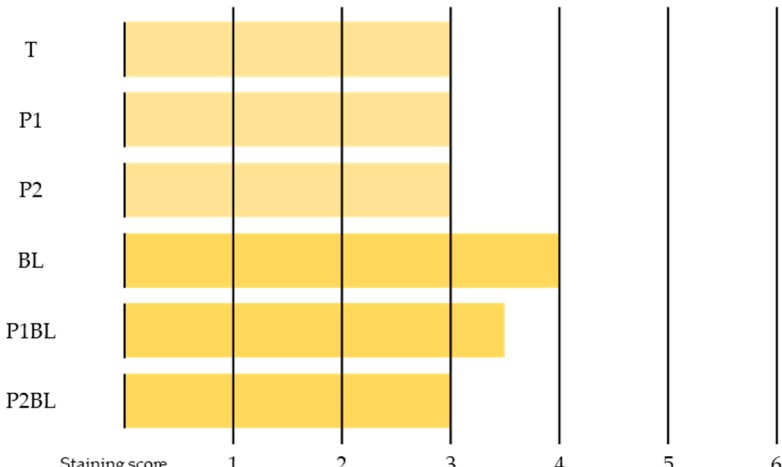

**Figure 5.** Scoring of the Nrf2 immunostaining by microscopical examination on D6, on unexposed or blue light-exposed batches (BL). T is the untreated control, P1 and P2 are the batches treated with the *Melissa officinalis* phytocomplex at 0.05% and 0.1%, respectively.

The blue light irradiation induces a moderate increase of the activated (nuclear) form of Nrf2 in the living epidermis. The *Melissa officinalis* phytocomplex application at 0.05% $w/v$ induces a slight decrease of the activated form of Nrf2 activation upon blue light exposure, so it partially reduces blue light-induced Nrf2 increase.

In parallel, the phytocomplex application at 0.1% $w/v$ completely inhibits the blue light-induced Nrf2 increase (Figure 6).

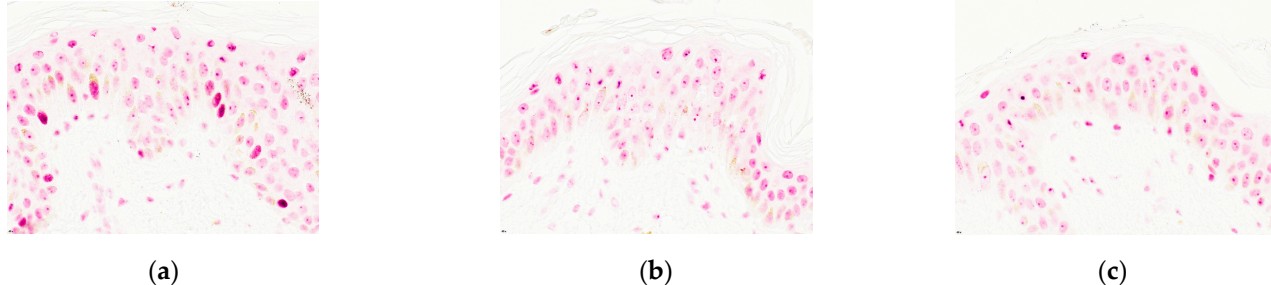

|       |       |       |
| :---: | :---: | :---: |
| (**a**) | (**b**) | (**c**) |

**Figure 6.** Nrf2 immunostaining of the batches irradiated with blue light (**a**), irradiated with blue light and pre-treated with the *Melissa officinalis* phytocomplex at 0.05% (**b**) or 0.1% *w/v* (**c**) on day 6.

### 3.3.3. Elastin Immunostaining

The results of the immunostaining of elastin in the papillary dermis on all the batches are summarized in Figure 7. After 6 days of treatment, the *Melissa officinalis* phytocomplex at 0.05 % *w/v* induces a slight decrease on elastin expression compared to the untreated control batch, while when applied at 0.1% *w/v*, no modifications are observed.

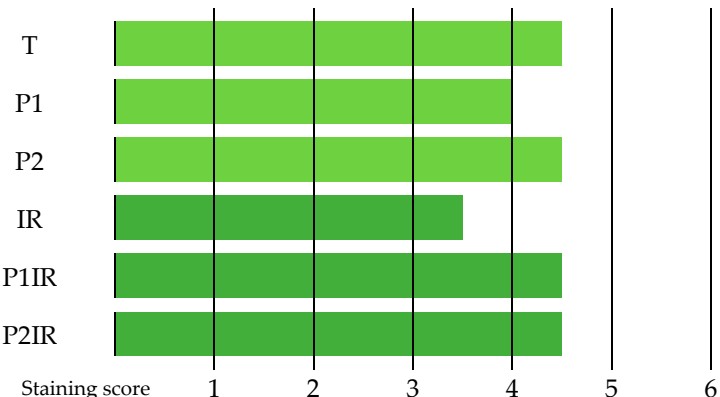

**Figure 7.** Scoring of the elastin immunostaining by microscopical examination on D6, on unexposed or infrared exposed batches (IR). T is the untreated control, P1 and P2 are the batches treated with the *Melissa officinalis* phytocomplex at 0.05% and 0.1% respectively.

The infrared irradiation induces a moderate decrease of elastin level in the papillary dermis. The *Melissa officinalis* phytocomplex applications at 0.05% *w/v* and at 0.1% *w/v* both induce a moderate increase on elastin level compared to the batch without the *Melissa officinalis* phytocomplex, suggesting that it totally inhibits IR-induced elastin degradation (Figure 8).

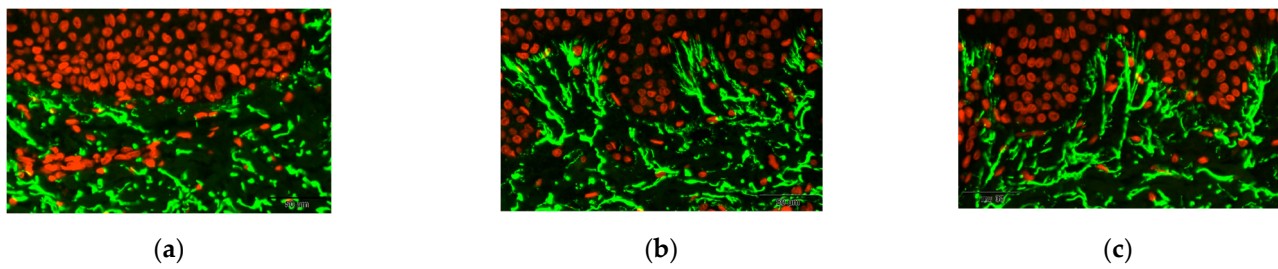

|       |       |       |
| :---: | :---: | :---: |
| (**a**) | (**b**) | (**c**) |

**Figure 8.** Elastin immunostaining in the papillary dermis of the batches irradiated with infrared on D6 (**a**), irradiated with infrared and pre-treated with the *Melissa officinalis* phytocomplex at 0.05% (**b**) or 0.1% *w/v* (**c**), on D6.

### 3.3.4. MMP-1 Immunostaining

The results of the staining of MMP-1 in the papillary dermis on all the batches are summarized in Figure 9. After 6 days of treatment, the *Melissa officinalis* phytocomplex 0.05% $w/v$ induces a slight increase of MMP-1 level, whereas when applied at 0.1% $w/v$, the phytocomplex induces a slight decrease of MMP-1 level compared to the untreated control batch.

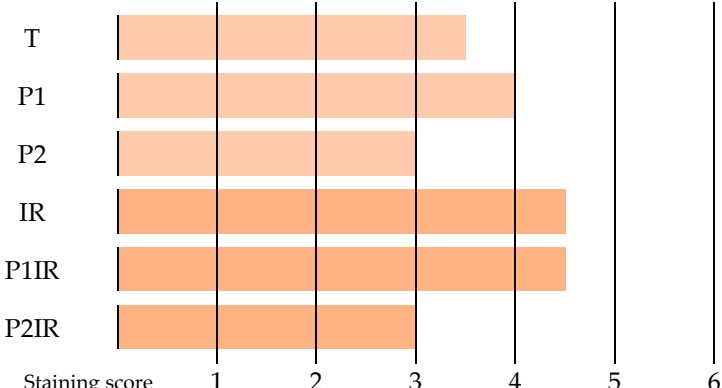

**Figure 9.** Scoring of the MMP-1 (Matrix Metalloproteinase 1) immunostaining by microscopical examination on day 6 on unexposed or infrared exposed batches (IR). T is the untreated control, P1 and P2 are the batches treated with the *Melissa officinalis* phytocomplex at 0.05% and 0.1% respectively.

Upon IR irradiation, the *Melissa officinalis* phytocomplex at 0.05% $w/v$ induces no variation of MMP-1 level in the papillary dermis. On the contrary, the phytocomplex at 0.1% $w/v$ induces a fairly clear decrease, so it totally prevents IR-induced MMP-1 increase in the dermis (Figure 10).

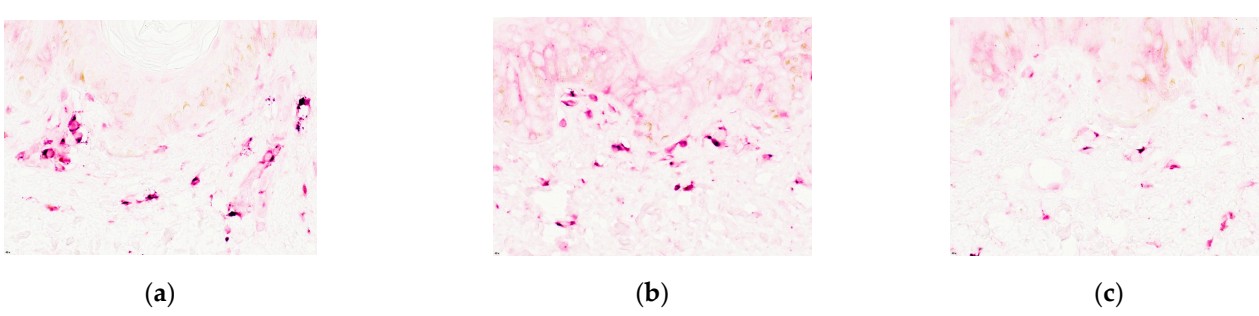

(**a**)                    (**b**)                    (**c**)

**Figure 10.** MMP-1 immunostaining in the papillary dermis of the batches irradiated with infrared on D6 (**a**), irradiated with infrared, and pre-treated with the *Melissa officinalis* phytocomplex at 0.05% (**b**) or 0.1% $w/v$ (**c**) on D6.

## 4. Discussion

The *Melissa officinalis* phytocomplex with a high and a standardized content of rosmarinic acid (7.6 ± 0.1% $w/w$) was obtained by using a plant cell culture technology [7]. Traditional plant extracts have extreme variability in the phytocomplex composition, and it depends on a lot of factors such as climate, soil, and cultivation techniques. This variability cannot guarantee the efficacy of the extract in health care applications. Using in vitro cell culture technology, we can obtain standardized products free from pesticides, contaminates, and residual solvents, maintaining similar biological efficacy in all batches [6]. The *Melissa officinalis* phytocomplex is standardized in the rosmarinic acid and total polyphenols content, which confers highly reproducible biological activities.

The use of the plant secondary metabolite rosmarinic acid has been studied in pharmaceutical and dietary supplements in Alzheimer's disease, atopic dermatitis, and cardio-

vascular disease [18,19]. Previous research has revealed that the effects of rosmarinic acid in these contexts are mediated by its antioxidant properties [20].

In the present work, we demonstrate the antioxidant properties of the *Melissa officinalis* phytocomplex using a stress oxidative model in the human cell line HaCaT after stimulus with $H_2O_2$. Keratinocytes represent 95% of the epidermal cells. Primarily, they play the structural and barrier function of the epidermis, but their role in the initiation and perpetuation of skin inflammatory and immunological responses, and wound repair, is also well recognized. The spontaneously immortalized human cell line HaCaT exhibits normal morphogenesis and expresses all the major surface markers and functional activities of isolated keratinocytes; for this reason, HaCaT will be chosen as a model for the study of keratinocytes functions and for investigating the effects of the compounds of interest on an epidermal model. Lorrio et al. demonstrated that a natural aqueous extract of *Deschampsia antarctica* protects skin against artificial ad natural light by reducing the production of ROS and hyperpigmentation in fibroblasts and melanocytes [11]. In this study, the chemical composition of the natural aqueous extract of *Deschampsia antarctica* was not described, while in our work, we obtained a phytocomplex highly strandardized in polyphenol content using the cell culture technology. The standardization in marker metabolites of the final product is a key point to guarantee the antioxidant efficacy of the *Melissa officinalis* phytocomplex.

Skin is the essential barrier protecting organisms against environmental insults and minimizing water loss from the body. Skin aging and dysfunctions are determined by several factors, including internal metabolism and environmental toxicant exposure. One of the leading causes of these processes is oxidative stress. Although there are efficient antioxidant systems in skin, the excessive and uncontrolled production of ROS is a major pathogenic factor that causes a range of skin diseases, inflammations, allergic reactions, and also neoplastic processes. To assess the ability of the *Melissa officinalis* phytocomplex to modulate the oxidative stresses, the general ROS levels in HaCaT cells were evaluated using a chemical fluorescent probe. The results obtained show that among a significant increase in ROS levels induced by oxidative stimulation, 0.1% *w/w* of the *Melissa officinalis* phytocomplex is able to prevent ROS formation, showing a potent antioxidant effect. The effect of the *Melissa officinalis* phytocomplex is comparable to the one observed for the pure compound Rutin (0.1% *w/w*) and is stronger than NAC (N-Acetyl-L-Cysteine) at 5 mM. This result indicates a potential protective effect of the *Melissa Officinalis* phytocomplex against ROS-mediated cell impairments.

Probably, the *Melissa officinalis* phytocomplex with a high content of rosmarinic acid (7.6% *w/w*) rescued $H_2O_2$-mediated inhibition of Nrf2 transcriptional activity and the consequential decrease in Nrf2 target gene expression [20]. Consistent with these findings, a previous study demonstrated that rosmarinic acid inhibits UVB-induced ROS production and the decrease in protein levels encoded by Nrf2 target genes in HaCaT keratinocytes [14].

Nrf2 is a key transcription factor in the cellular response to oxidative stress. Human Nrf2 has a predicted molecular mass of 66 kDa, and it is ubiquitously expressed in a wide range of tissues and cell types. Under oxidative stresses, including UV irradiation, Nrf2 is activated by phosphorylation and translocates from the cytoplasm to the nucleus. So far, different cytosolic kinase, including protein kinase C (PKC), phosphatidylinositol 3-kinase (PI3K), mitogen-activated protein kinase (MAPK), andER-localizedpancreatic endoplasmic reticulum kinase (PERK) have been shown to modify Nrf2 and are potentially involved in the dissociation of Nrf2 from its inhibitor, Keap1 [21]. Once in the nucleus, Nrf2 binds to the DNA at the location of the Antioxidant Response Element (ARE) or also called hARE (Human Antioxidant Response Element), which is the master regulator of the total antioxidant system. Nrf2 plays a role in protecting human skin keratinocytes from UVA radiation-induced damage [22] and environmental pollution [23]. The protection effect of the *Melissa officinalis* phytocomplex against the irradiation of blue light was evaluated on living human skin explants ex vivo by the immunostaining of Nrf2. As shown in Figure 6, the blue light irradiation induces an increase of nuclear translocation of Nrf2

in the living epidermis. The *Melissa officinalis* phytocomplex at 0.05% $w/w$ presents a fairly good protection activity against blue light irradiations by partially reducing the Nrf2 blue light-induced activation. The higher concentration (0.1% $w/w$) of the phytocomplex presents a good protection activity against blue light irradiations by totally reducing the Nrf2 blue light-induced activation, suggesting that the *Melissa officinalis* phytocomplex is able to maintain a healthy balance of anti-oxidant response upon blue light irradiation.

The protection effect of the *Melissa officinalis* phytocomplex against the irradiation of infrared was evaluated on living human skin explants ex vivo by immunostaining of elastin and MMP-1.

Elastin is a protein of the connective tissue responsible for the elastic properties of the skin. It is rich in hydrophobic amino acids such as glycine and proline, which form mobile hydrophobic regions bounded by crosslinks between lysine residues, and it is localized mainly on the elastic fibers of the papillary dermis [24]. The principal function of elastin is its ability to elastically extend and contract in repetitive motion when hydrated. Elastin is the major component (90%) of the elastic fibers of the skin. Several factors including the formation of methylglyoxal [25] or UV exposure are able to damage the elastic network of dermis, including elastin [26]. In dermo-cosmetic research, elastin is usually used as a bio-marker of the structural state of the extracellular matrix of the dermis.

The matrix metalloproteinases (MMPs) are a family of peptidase enzymes responsible for the degradation of extracellular matrix components, including collagen I, gelatin, fibronectin, laminin and proteoglycan. However, it also has other substrates such as Pro-MMP-1, Pro-MMP-2, proMMP-9, MCP (Monocyte Chemoattractant Protein)-1, MCP-3, MCP-4, Stromal Cell-Derived Factor (SDF), Pro-1L-1β, 1L-1β, IL-8, IGFBP-2 (Insulin-like growth factor-binding protein 2), IGFBP-3 (Insulin-like growth factor-binding protein 3), Pro-TNF-$\alpha$ (Tumor necrosis factor-$\alpha$), CXCL5, CXCL11 precursor, Casein, C1q, Serum amyloid A protein, $\alpha$1-Proteinase Inhibitor, $\alpha$1- Anti-Chymotrypsin, $\alpha$2-Macroglobulin.

MMP-1 is an enzyme also known as interstitial collagenase and fibroblast collagenase. MMP-1 is expressed in migrating keratinocytes via ligation of the $\alpha$2β1 integrin with dermal collagen and is a reliable marker of activated keratinocytes in wounded human skin. Other factors, including smoke [27] and UV exposure [28], induce MMP-1 expression, leading to premature skin aging.

The *Melissa officinalis* phytocomplex at 0.05% $w/w$ shows a moderate protection activity against infrared irradiations by totally preventing the IR-induced elastin alteration (Figure 8). The higher concentration (0.1%$w/w$) of the phytocomplex shows a good protection activity against infrared irradiations by totally inhibiting the elastin alteration and MMP1 release induced by IR exposure (Figure 10). These results indicate that the *Melissa officinalis* phytocomplex at 0.1% $w/w$ may protect the skin against the damages of oxidative stress and blue light by reduction of Nrf2 activation. In addition, the product protects the skin against the infrared irradiations damage by the inhibition of elastin alteration and MMP-1 release.

## 5. Conclusions

In conclusion, the findings of the present study suggested that the *Melissa officinalis* phytocomplex is a new standardized cosmetic ingredient obtained by an in vitro plant cell culture with a high effectiveness to protect skin against oxidative stress, blue light, and irradiations of infrared damages. The in vitro plant cell culture has allowed us to obtain a phytocomplex with high content of rosmarinic acid and without environmental contaminants using an eco-sustainable process. This *Melissa officinalis* phytocomplex demonstrate antioxidant activity by reducing ROS production and thus the oxidant damage of the skin caused by UV and blue light exposure. In addition, the efficacy of this phytocomplex is related to the inhibition of blue light-induced Nrf2 transcriptional activity, IR-induced elastin alteration, and IR-induced MMP-1 release. The *Melissa officinalis* phytocomplex is a new innovative active ingredient for cosmetic products that is able to protect skin against light and screen exposure damages.

## 6. Patents

Patent ITA102019000004113-PCT/IB2020/052589: Phytocomplex and selected extract of a meristematic cell line of a plant belonging to the genus Melissa.

**Author Contributions:** Conceptualization, G.P., O.B., A.S. and F.G.; methodology, G.P., O.B., C.G., F.G. and C.D.; validation, O.B. and C.G.; formal analysis, V.C., I.G. and L.P.-M.; investigation, G.P., O.B. and L.P.-M.; writing—original draft preparation, G.P. and E.B.; writing—review and editing, G.P., F.G. and E.B.; supervision, G.P. All authors have read and agreed to the published version of the manuscript.

**Funding:** This research received no external funding.

**Institutional Review Board Statement:** The ex vivo study is performed in full respect with the Declaration of Helsinki and the article L.1243-4 of the French Public Health Code. The latter does not require any prior authorization by an ethics committee for sampling and using surgical wastes.

**Informed Consent Statement:** Informed consent was obtained from all subjects involved in the study.

**Acknowledgments:** Thanks to all technical team of Demethra biotech Srl for the support in the development of Melissa officinalis cell line.

**Conflicts of Interest:** The authors declare no conflict of interest.

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
