# Peer review of "In Vitro Cultured Melissa officinalis Cells as Effective Ingredient to Protect Skin against Oxidative Stress, Blue Light, and Infrared Irradiations Damages"

_cosmetics, doi:10.3390/cosmetics8010023_

Round 1
Reviewer 1 Report
The article is an interesting original study on the use of "melissa Officinalis Phyto complex " to protect skin against blue light and infrared damages, evaluating the ROS level in a keratinocyte cell line from human skin (HaCaT) and Nrf2, Elastin, and MMP1 immunostaining in living human skin explants ex vivo.
Although the article is very interesting, it needs a round of review:
No sub-section in the material and methods section describes how the statistical analysis was performed. It is crucial to indicate what types of tests were used to calculate significance, what statistical program was used (and also the company that makes it), and if any other tests or corrections for comparisons were used.
Page 2 line 72-74 "Sunlight spectrum consists of short, high-energy wavelengths, from ultraviolet radiation (280–400 nm) to visible light (400–700 nm) d infrared radiation (700 nm–1 mm), long and low energy wavelengths."This phrase needs a reference: I suggest you this one doi: 10.1007/s10103-020-03063-6.
Page 12 line 447-451 "Elastin is a protein of the connective tissue responsible for the elastic properties of the skin. It is rich in hydrophobic amino acids such as glycine and proline, which form mobile hydrophobic regions bounded by crosslinks between lysine residues and it localized mainly on the elastic fibers of the papillary dermis. The principal function of elastin is its ability to elastically extend and contract in repetitive motion when hydrated. Elastin is the major component (90%) of the elastic fibers of the skin. " This whole period needs a reference; I suggest you this one: doi: 10.1089/photob.2020.4908.
Thank You
Author Response
Dear Reviewer,
thank you for appreciatiating our work.
We add the sub-section in the material and methods 2.6 called statistical analysis where we reported the statistical program that have been used.
Page 2 line 72-74 we add the reference doi:10.3390/ijms21218020 because we think that it reports exactly the subject matter.
Page 12 line 455 we add the reference doi: 10.1111/ics.12372 because we think that it reports exactly the subject matter.
Please see attached reviewed manuscript with all reviewers corrections in different colours.
You can find your revisions highlighted in yellow.
Thank you and kind regards

Reviewer 2 Report
The manuscript by Pressi et al., describes an interesting approach to control the problematic of variation of plant constituents over time, which may hinder the reprodutibility of cosmetic formulations.
The introduction provides sufficient background, the research design is appropriate and the work is very interesting for the readers of this journal. However, some details should be improved, namely:
- the statement in lines 40-47 should be supported by a reference;
- In line 85, where it reads "were shown" should be corrected to "showed";
- in line 121 "was transfer" should be "was transferred";
- line 131 - "the suspensions" should be "the suspension";
- line 134 - species name must be in italic
- line 149 - the sentence "The Melissa officinalis phytocomplex was extracted in fifth fold." is confusing. Fifth fold compared with what? Please clarify;
- Lines 252-254 - the constitution of the medium was explained in the material and methods section. It does not need to be stated again in the Results section;
- Figure 3 - the dark bar should be labeled "Control";
- Figure 8 - there is a formating problem that prevents a correct interpretation of the figure;
- Line 380-383 - this sentence should be supported by a reference;
- The back matter is not completed and the information about the Informed Consent Statement must be added to this section, as stated in the Cosmetics journal instructions for authors
Author Response
Dear Reviewer,
thank you for appreciating our work.
We correct all the points of your revision as you suggest:
The statement in lines 40-47 should be supported by a reference - we add the reference number 1 https://doi.org/10.3390/pr7020088 and another reference DOI: 10.1177/2156587216663433
In line 85, where it reads "were shown" should be corrected to "showed"- corrected
in line 121 "was transfer" should be "was transferred"; - corrected
line 131 - "the suspensions" should be "the suspension"; - corrected
line 134 - species name must be in italic - corrected
line 149 - the sentence "The Melissa officinalis phytocomplex was extracted in fifth fold." is confusing. Fifth fold compared with what? Please clarify; - corrected
Lines 252-254 - the constitution of the medium was explained in the material and methods section. It does not need to be stated again in the Results section; - delated
Figure 3 - the dark bar should be labeled "Control"; - corrected
Figure 8 - there is a formating problem that prevents a correct interpretation of the figure; - corrected
Line 380-383 - this sentence should be supported by a reference; - we add the reference number 5
The back matter is not completed and the information about the Informed Consent Statement must be added to this section, as stated in the Cosmetics journal instructions for authors - we complited the sections about founding, conflicts of interest and acknowledgments
Please see the attached reviewed manuscript with all reviewers corrections in different colours.
You can find your revisions highlighted in green
Thank you and kind regards

Reviewer 3 Report
I thing the manuscript entitled "In vitro cultured Melissa officinalis cells as effective ingredient to protect skin against oxidative stress, blue light and infrared irradiations damages" really interesting. The article presents innovative results of research in the aspect of action of the active ingredients of Melissa officinalis.
The manuscript is well-organised, methods and results are clearly and carefully described. In the article many studies have been conducted that confirm the protective effect of Mellisa on the skin. I strongly reccomend this manuscript for publication. The results will be very useful both for other researcher and for employees work for R&D departments in chemical, farmaceuthical and cosmetics companies.
I have only one suggestion for authors. In the discussion section authors should discuss their results more with results previously obtained by another authors and not mainly with theoretical aspects.
Also, I think that introduction could be a little shorter.
Author Response
Dear Reviewer,
thank you for appretiating our work
In the discussion we compared our work to one about another cosmetic active ingredients called Edafence because it is the only article we found about natural raw materials and blue light activity
We think that the introduction could be a little shorter but we introduced all the arguments treated in the article so we hope you understand if we don't shorten it.
Please see the attached reviewed manuscript with all reviewers corrections in different colours.
You can find your revisions highlighted in light blue
Thank you and kind regards

Round 2
Reviewer 1 Report
The authors responded to all queries. The paper is publishable.